# Using an analogue-digital hybrid clinical data management platform during a two-dose preventive Ebola virus vaccine trial in Goma, the Democratic Republic of the Congo

Hannah E. Brindle[1]*, Darius Tetsa-Tata[1], Tansy Edwards[1], Edward Man-Lik Choi[1], Kambale Kasonia[1], Soumah Aboubacar[2], Grace Mambula[2], Hugo Kavunga-Membo[3], Rebecca Grais[2], John Johnson[4], Daniel G. Bausch[1,5], Jean-Jacques Muyembe-Tamfum[3], Ibrahim Seyni Ama[2], Shelley Lees[1], Deborah Watson-Jones[1,6], Anton Camacho[2], Chrissy H. Roberts[1]*

1 London School of Hygiene & Tropical Medicine (LSHTM), London, United Kingdom, 2 Epicentre, Paris, France, 3 Institut National pour la Recherche Biomédicale (INRB), Kinshasa, DR Congo, 4 Médecins sans Frontières, Paris, France, 5 FIND, Geneva, Switzerland, 6 Mwanza Intervention Trials Unit, National Institute for Medical Research, Mwanza, Tanzania

☯ These authors contributed equally to this work.
* hannah.brindle@lshtm.ac.uk (HEB); chrissy.roberts@lshtm.ac.uk (ChR)

## Abstract

Clinical trials in settings with intermittent or non-existent internet and power connectivity, for example during humanitarian emergencies, present challenges in the synchronisation of data across different sites, in addition to accessing a centralised database in real-time. To overcome these, we designed a novel hybrid analogue/digital data management system which was deployed during the rapid implementation of a Phase III evaluation of a two-dose preventative vaccine for Ebola virus disease in Goma, Democratic Republic of the Congo, from 2019 to 2022. We provided study participants with an Enhanced Participant Record Card (EPRC) that served as eligibility for, and confirmation of, vaccination and was used in combination with Open Data Kit (ODK) electronic case report forms to create an off-grid study participant management system. To understand the utility of the EPRC, we analysed data from 15,327 study participants who received both vaccines and various types of prompts or reminders to return for dose 2, including home visits, telephone calls, or short messaging service (SMS). A total of 53% participants referred to the date on the EPRC as a prompt to return for dose 2 and 36.1% mentioned this as the only prompt. A multivariable generalised linear mixed-effects model showed that those who were not working, those aged 45–64 years or who had a chronic medical condition identified prior to receiving dose 2 were more likely to use the date on the EPRC as a prompt. Our findings demonstrate the utility of this system in the facilitation of decentralised data collection in off-grid locations that may be useful for future trials in complex humanitarian settings.

**Clinical Trials Registration Number:** ClinicalTrials.gov NCT01128790

**Data availability statement:** The rights of study subjects and partners, the sharing of data between partners, and the transfer of data to external third parties are governed by the Data Sharing Agreement. Aggregated data are available at clinicaltrials.gov. De-identified participant-level data collected in this trial will be disseminated through the LSHTM Data Compass, a FAIRcompliant data repository (https://datacompass.lshtm.ac.uk/. Any requests for de-identified data prior to these being made publicly available, can be made to the chair of the observational ethics committee at LSHTM (ethics@lshtm.ac.uk). Additionally, the R code for analyses can be made available from the same address.

**Funding:** This study was supported through funds from the Coalition for Epidemic Preparedness Innovations (CEPI) [FELS1903] and the Paul G. Allen Family Foundation. This work was also supported by the UK Foreign, Commonwealth & Development Office (FCDO) and Wellcome [220506/Z/20/Z] and by the European Union's Horizon 2020 research and innovation programme under grant agreement No 857935. Chrissy h Roberts was funded by the Department of Health and Social Care using UK Aid Funding as part of the UK Vaccine Network, which is managed by the National Institute for Health and Care Research (PR-OD-1017-20001). The views expressed in this publication are those of the authors and not necessarily those of the Department of Health and Social Care, CEPI, the Paul G. Allen Family Foundation, FCDO, Wellcome or the European Commission. The funders had no role in study design, data collection and analysis, decision to publish, or preparation of the manuscript. CEPI provided funding as well as technical support for the study. However, all final decisions regarding data analyses, presentation and publication were the responsibility of the Principle Investigator from INRB and Co-Principle Investigators from LSHTM.

**Competing interests:** The authors have declared that no competing interests exist.

## Introduction

Robust data management methods must be embedded aspects of clinical trials and human participant research [1]. The longitudinal nature of most clinical trials, including trials with repeated interventions, necessitates the use of a data system that can manage records of information about trial participants across time. If used according to Good Clinical Practice (GCP) standards [2], a trial's clinical data management system (CDMS) forms the core of both logistic and scientific trial activities. It ensures tracking and recording of adherence with the study protocol, safety of the participant, maintaining the quality of the data collected, and providing evidence that a trial adhered to the ethical, scientific and practical standards of GCP.

In the simplest instance, a trial's CDMS might take the form of an analogue paper registry or filing system but, whilst paper records still have a key role in some aspects of trial documentation (i.e., participant identification, case report forms and medical records), data are typically transferred to a centralised digital database. The majority of modern CDMSs are web-based and rely on structured query language databases [3]. This is advantageous because these provide fixed data structures and an auditable trail of data entry and modification. Web-based systems are also ideal for studies that are spread across different geographical sites since they can synchronise data in real-time between workstations at study locations, clinics, pharmacies, laboratories, coordinating centres, and to monitors, sponsors, and research partners. CDMSs are typically complex multi-component platforms that use a plurality of information technology solutions to bring together all information that is relevant to the documentation of a trial's activities. A key use of the CDMS is in the logistic delivery of a trial, specifically in supporting the effective management of participants as they progress through the various stages of a trial protocol. No consistent naming convention exists that is able to describe the various components of the archetypal CDMS [1], but we refer here to the Study Participant Management System (SPMS) as the components of the CDMS that facilitate the recruitment, follow-up, adherence with the study intervention, and management of individual participants, as well as the primary entry of longitudinal data about those individuals.

Epidemics and pandemics present an urgent need to conduct clinical trials of novel medical countermeasures, but such events frequently occur in areas where web-based electronic CDMS platforms are not a reliable solution. This is because the range of available options for effective electronic clinical data management becomes more limited when access to the worldwide web, electricity, information technology, and general supporting infrastructure is unreliable, inconsistent or non-existent [4]. During the 2014–2016 West Africa Ebola epidemic, Médecins sans Frontières (MSF), in collaboration with software developers, produced an electronic data-capture system where internet connectivity in Ebola management centres in Sierra Leone was intermittent and where there was a risk of transmission of the Ebola virus via electronic equipment. The bespoke interface was based on elements from OpenMRS and Open Data Kit (ODK). Data were entered into a custom-built Android application installed on Sony Android tablets which could be disinfected in chlorine. Although there was a requirement for a local server to synchronise data between the tablets and for data to be stored outside the high-risk zone, data were backed up to the server using USB sticks rather than the internet. Following this study, the authors recommended a framework for innovative technology projects in humanitarian settings including developing high-level and implementation requirements as well as an evaluation plan [5]. During the same epidemic, the Clinical Trials Unit (CTU) Bern developed an electronic clinical trial data management system for two Ebola vaccine trials in Conakry, Guinea. Although the team encountered similar challenges with IT infrastructure including frequent power cuts, the team members also needed to collect longitudinal data. For this purpose, they use the web application, Research Electronic Data Capture (REDCap). The server comprised a MacBook computer connected to a network switch and a

WiFi router. Although the system worked well, the authors noted that it was not fully compliant with data management standards including validation [6]. Following the 2014 Ebola epidemic in Boende district, western DRC, a vaccine trial of the 2-dose heterologous Ad26. ZEBOV, MVA-BN-Filo regimen (EBL2007) took place between 2019 and 2022. However, again due to limited internet connectivity, a local server was used for data entry with these copied over on a daily basis to a central server using a satellite uplink [7].

Between 2018 and 2020, an outbreak of Ebola virus disease (EVD) occurred in the eastern region of the Democratic Republic of the Congo (DRC), with 3,470 confirmed cases and 2,287 deaths [8]. During this outbreak, a consortium of national and international global health actors undertook a Phase III non-randomised, open-label, single arm evaluation of the effectiveness, safety and immunogenicity of a heterologous two dose (Ad26.ZEBOV, MVA-BN-Filo) preventative EVD vaccine regimen (the DRC-EB-001 trial) [9,10]. The study was conducted in Goma, the capital of North Kivu Province, between 14/11/2019 and 9/2/2021, with trial activities paused between 10/4/2020 and 15/9/2020 due to the COVID-19 pandemic. Internet and power availability is limited and inconsistent in Goma, posing challenges to a web-based SPMS. In this paper, we describe a novel off-grid SPMS which employed a hybrid analogue-digital solution for a two-dose vaccine trial in a complex setting and the utility of the EPRC as a prompt to return for the second vaccine dose. Although the first aspect draws on similar CDMS approaches deployed during the West Africa Ebola epidemic, the integration of this with an EPRC is a novel feature.

## Materials and Methods

### Ethics, trial and study design

The trial was approved by the London School of Hygiene & Tropical Medicine research Ethics Committee (ref. 17471), the Médecins sans Frontières (MSF) Ethics Review Board (ref. 1922), the DRC Avis du Comité National d'Ethique de la Santé (ref. 140/CNES/BN/PMMF/2019) and Comité d'Ethique, Université de Kinshasa (ESP/CE/250/2019). Full details of the trial design, information about informed consent and assent processes, criteria defining participant eligibility and exclusion, and further information about quality assurance of the trial have been previously described [10,11]. In summary, a test-negative design was proposed to determine vaccine efficacy with safety, coverage and knowledge and perceptions of the vaccines included as secondary objectives [10]. This descriptive sub-study used data available from all participants who received their second dose.

### Study site

Goma, a city of approximately 1.5 million people, is located on the busy border between DRC and Rwanda and is divided administratively into the Goma and Karisimbi Health Zones, which are further divided into Health Areas. The trial, which operated in six vaccination sites located in Majengo and Kahembe, two of the seventeen Health Areas in Karisimbi. These Health Areas were chosen due to their links with the cities of Beni and Butembo which were the epicentre of the epidemic and therefore, the potential for transmission of the virus among the population [10]. All adults and children aged one year and over were invited to participate in the trial if they lived or worked in the communities and planned to remain there for one month following dose 2 [10]. Whilst participants were encouraged to consistently visit the same site throughout the study, this was not always the case, particularly if the participant needed to report a serious adverse event (SAE) or other important medical event, such as a pregnancy or childbirth. If participants were unwell, they were able to present at any study site, at local healthcare centres or at any Ebola Treatment Centre (ETC) (if they had

symptoms suggestive of EVD) within the region. Pregnant women were also able to deliver at any hospital within Goma but were encouraged to use hospitals assigned by MSF, one of the study partners.

Internet connectivity and electrical supply at the study sites was inconsistent. Access to the web was fully dependent on access to 3G or 4G mobile connections. Furthermore, due to security concerns in Goma, it was considered unsafe to use computers at study sites due to the risk of being stolen. Instead, mobile devices (tablets) were deployed for recording consent and participant data since they were small, portable, discreet and provided data security through encryption. In addition, tablets routinely provide 8–12 hours of battery life between charges, meaning that they could be relied on to last a full working day without access to electricity. They were returned to the study project office for data downloading and charging at the end of each workday and where they could be kept safely.

## Specification and design of the clinical data management system

In the absence of a web-based data management system at the vaccination sites, the study required tools that could [i] be used to assign unique study identification (ID) numbers to participants and to provide a simple device by which a participant's study ID could be accurately and longitudinally referenced on all relevant case reporting forms (CRFs) in order to link them together, [ii] confirm that the individual was the same person seen at a previous visit(s), [iii] assist in managing the participant flow at vaccination sites, [iv] allow deferment or exclusion of ineligible and/or contraindicated participants, [v] remind the participant of the dates of future vaccination and study visits, [vi] provide information required for safety monitoring (i.e., by providing information on how to contact the study team), [vii] provide documentation of study participation to a healthcare provider or an ETC if needed and [viii] comply with study protocols related to data protection.

This specification was subject to several conditions; the system was required to work without the need for a reliable internet connection or constant power supply and without real-time data synchronisation between local devices and/or a central database. It also needed to be able to record when participants presented at different study sites or clinics to allow staff at the vaccination sites, healthcare centres or ETCs to identify and monitor a participant's progression without having to locate paper records. Furthermore, it was crucial that the system could clearly differentiate between participants who were, according to the study protocol, ineligible for further vaccination either temporarily (for example, feeling unwell on the day) or permanently (for example, because of a previous SAE related to the vaccine).

Daily operations at the vaccination sites were centred around a queue-based system (Fig 1) in which participants physically moved between 'stations' dedicated to individual and sequential tasks. At each of the two visits to a study site for vaccination with either Ad26.ZEBOV ('dose 1' at day 0) or MVA-BN-Filo ('dose 2' at day 56 {-14/ +28}, prior to the suspension of activities due to the COVID-19 pandemic, when the second dose was often delayed by necessity).

As participants progressed through visit procedures, they were assessed for eligibility, offered a pregnancy test (females whose last menstrual period was more than 28 days ago or who were unsure if they were pregnant) and, if eligible, vaccination, and then observed for 15 minutes post-vaccination before departure (Fig 1). Pregnancy testing was optional given there is no evidence that the vaccines are teratogenic and therefore pregnant women were not excluded from vaccination. However, women who were pregnant at the time of the dose 1–30 days after dose 2 were followed up for three months after delivery to obtain safety data. Additionally, 360 pregnant women were actively followed up during pregnancy to collect data on serious adverse events (SAEs) [11]. All study participants were passively monitored for SAEs

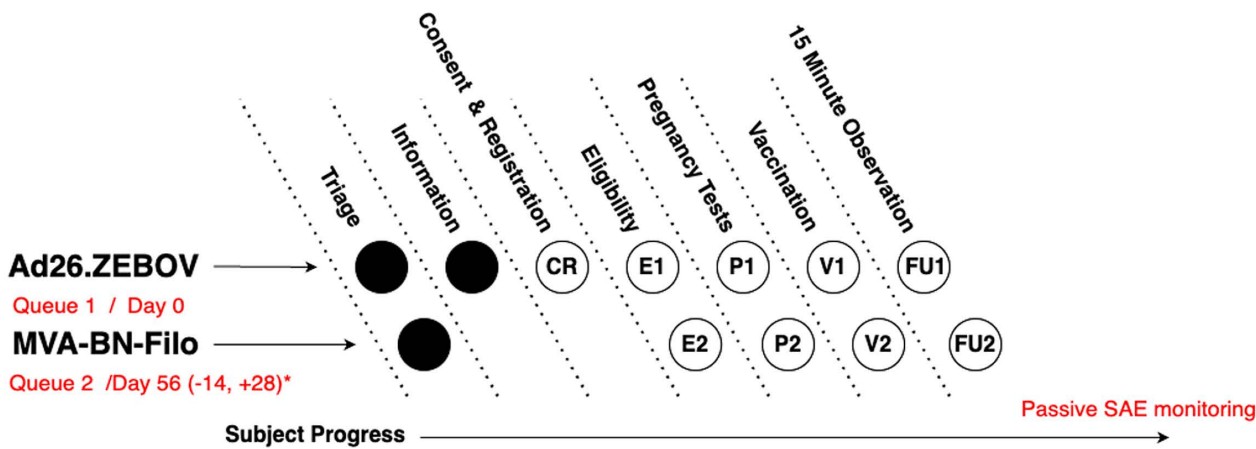

**Fig 1. Participant progression through the vaccination sites for dose 1 and dose 2.** Vaccination processes at the study site took place within a two-queue system. Queue 1 was for Ad26.ZEBOV vaccination (dose 1) and Queue 2 for MVA-BN-Filo (dose 2). Participants entered Queue 2 at any time between days 42 and 84 (ideally on day 56, see main text). Within each queue, participants moved between stations in the manner of a production-line. Study staff at each station performed specific tasks, using the Enhanced Participant Record Card to identify the participant, manage their movement around the site and record data via the ODK data collection system (see main text below). Circles indicate stations visited by participants in each queue. Open circles indicate stations where data entry with ODK occurred. Codes in open circles map to data tables in the relational database.

(i.e., instructed to contact the study team, return to the study sites or present to a healthcare provider with any health problems or concerns) or actively followed up for SAEs if they were part of a safety subset.

### The Enhanced Participant Record Card

During the consent and registration process, each participant was provided with a personalised EPRC. The EPRC contained information typically found on a vaccination card (name, age, date of vaccination, 24-hour emergency hotline), a photograph, plus additional features that were designed to address the needs of the specification of the CDMS described above. Pre-generated stickers were used to assign unique study IDs (a task usually performed by the SPMS database) (Fig 2). The EPRC allowed the participant to carry sufficient information about their study participation so that the study team at any site could safely and appropriately support them at any scheduled or unscheduled visits.

Staff at the consent and recruitment station initiated the process of EPRC construction by first taking a credit-card sized 46 mm × 62 mm (1.8 in × 2.4 in) polaroid photograph (INSTAX, FujiFilm) of the participant (and in the case of children, of the participant with their parent or guardian). A pre-printed sticker was affixed to the photograph (Fig 2). The sticker featured a unique study ID number in both human- and machine-readable (Aztec code) formats, along with several coded boxes which could be perforated by study staff as the participant progressed through the study visits and procedures. Identical duplicate stickers were then attached to each copy of the paper informed consent or assent form, one of which was retained by the participant and the other archived by the study team. The stickers attached to the EPRC, and consent/assent forms were the primary mechanism by which study IDs (*UniqueID*) were assigned to study participants and by which consent/assent forms could be linked back to the digital database in the future.

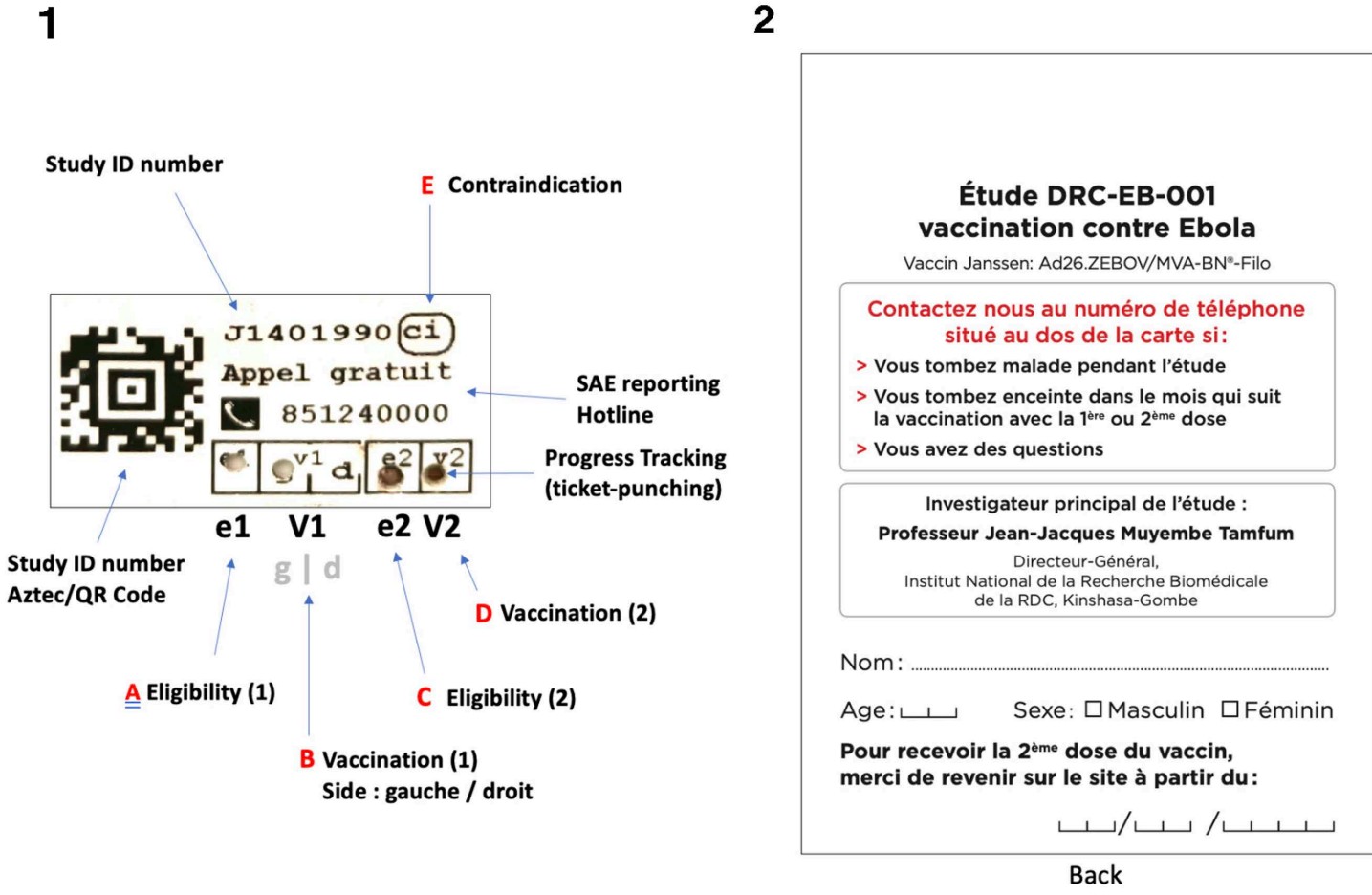

**Fig 2. An example of the components of the Enhanced Participant Record Card (EPRC) used in the study.** On the front, a sticker (1) was affixed to an instant photograph of the participant. On the back, the EPRC provided information about the study (2), details of the participant's identity and instructions to return to the site for the 2nd dose from a date calculated by the ODK data management system. The EPRC sticker (see detailed view, right of figure) provided mechanisms for off-grid participant management and logging of (A) eligibility for dose 1, (B) vaccination with dose 1 in the left ('gauche (g)') or right ('droit' (d)) arm, (C) eligibility for dose 2, (D) vaccination with dose 2 and (E) contraindication[s] preventing further participation in the study. Perforation of the boxes A-E using a ticket-punch provided a permanent and unmodifiable record on the EPRC. The information on this example is illustrative and does not relate to any real study participant.

In order to help study staff to manage the participants as they moved through each of the study sites, the EPRC provided an analogue mechanism by which various perforations to boxes printed on the sticker (Fig 2) could indicate the participant's progress to date, for instance, by showing which dose of vaccine they had received, or whether they were eligible for dose 2 but had yet to receive it. Study staff were equipped with 1.5mm ticket-punching tools and used these to perforate the indicated boxes on the EPRC when instructed to do so by specific electronic Standard Operating Procedures (eSOPs) and in response to specific data requirements having been met. At various points, eSOPs requested that field staff should check and confirm which boxes were perforated on a participant's EPRC (Figs 2–4), with progress through a form being halted when an incorrect configuration of perforations was identified. If a participant was eligible and received dose 1, the assembly of the EPRC was completed by writing the participant's name, age, sex and date to return for dose 2 on a pre-printed plastic vaccination card (85.6 x 53.98mm) (Fig 2) which was then sealed back-to-back with the sticker/photograph using a self-sealing, transparent plastic envelope.

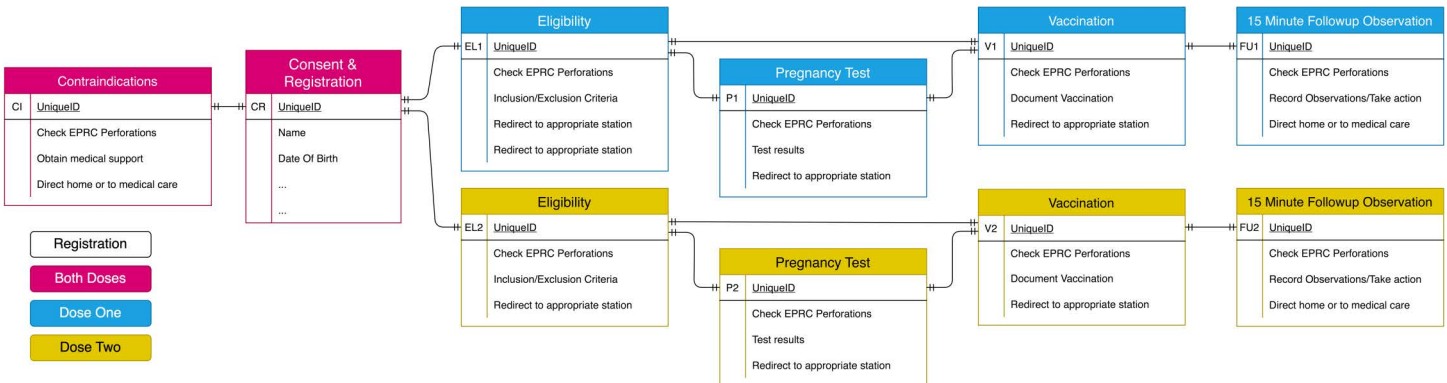

**Fig 3. Key actions performed at each station at the vaccination sites for doses 1 and 2.** The digital database schema was characterised by simple one-to-one relationships, with the consent and registration electronic case reporting forms (eCRF) comprising the key record of the participant entity, and the UniqueID field creating the link between the eCRFs.

## Off-grid electronic case reporting forms and standard operating procedures

We chose to use the electronic data collection software ODK as the basis of the digital database and SPMS. This builds on developments made during our previous study of another EVD vaccine, rVSV-ZEBOV-GP, also used during the 2018–2020 EVD outbreak in the DRC [12]. A key feature of ODK is that it can work entirely off-grid, uploading data to a server *ad hoc* when internet connections become available. An important limitation of ODK is that it provides limited native off-grid features that facilitate longitudinal and relational data structures to describe and track entities followed across time. As such, it was necessary to manage longitudinal data at vaccination sites as a plurality of separate, cross-sectional electronic CRFs (eCRFs) relating to a specific participant. These eCRFs were completed at the stations for consent and registration, eligibility, pregnancy testing, vaccination and the 15-minute observation period following vaccination. By including a common data field (for instance, the study ID number, here using the field *UniqueID*) in every form submission, it was possible to bind a participant's various eCRFs together *post hoc in silico* in addition to collecting further data required for the study such as whether the eligibility criteria were met, reasons for exclusion, pregnancy test results, reasons why a vaccine was not administered and SAEs reported during the observation period (Fig 3). The underlying design of ODK forms is the ODK xForm Specification [13], a format that serves simultaneously as an eCRF (i.e., an instrument to record study data) and an eSOP (an instrument to direct enumerators to follow protocols). Both eCRF and eSOP functions leverage ODK's sophisticated system of form logic, routing, and constraints, allowing the team to automatically assess participant data, and apply algorithms to, for example, assess eligibility criteria, or to determine which station a participant should be sent to next. Other examples of eSOP elements (Fig 4) included instructing the study team to ask a temporarily ineligible participant to return to the eligibility station three days later, directing a participant to progress to the next station, or offering a 20-year-old female whose last menstrual period was more than 28 days ago a pregnancy test prior to vaccination. All ODK forms used in the DRC-EB-001 trial functioned in French and English, with users being able to switch between languages as needed.

The ODK Collect application was installed on password-protected Android tablets and the completed, encrypted eCRFs were automatically synchronised to an ODK Aggregate server hosted at the London School of Hygiene & Tropical Medicine once WiFi was available

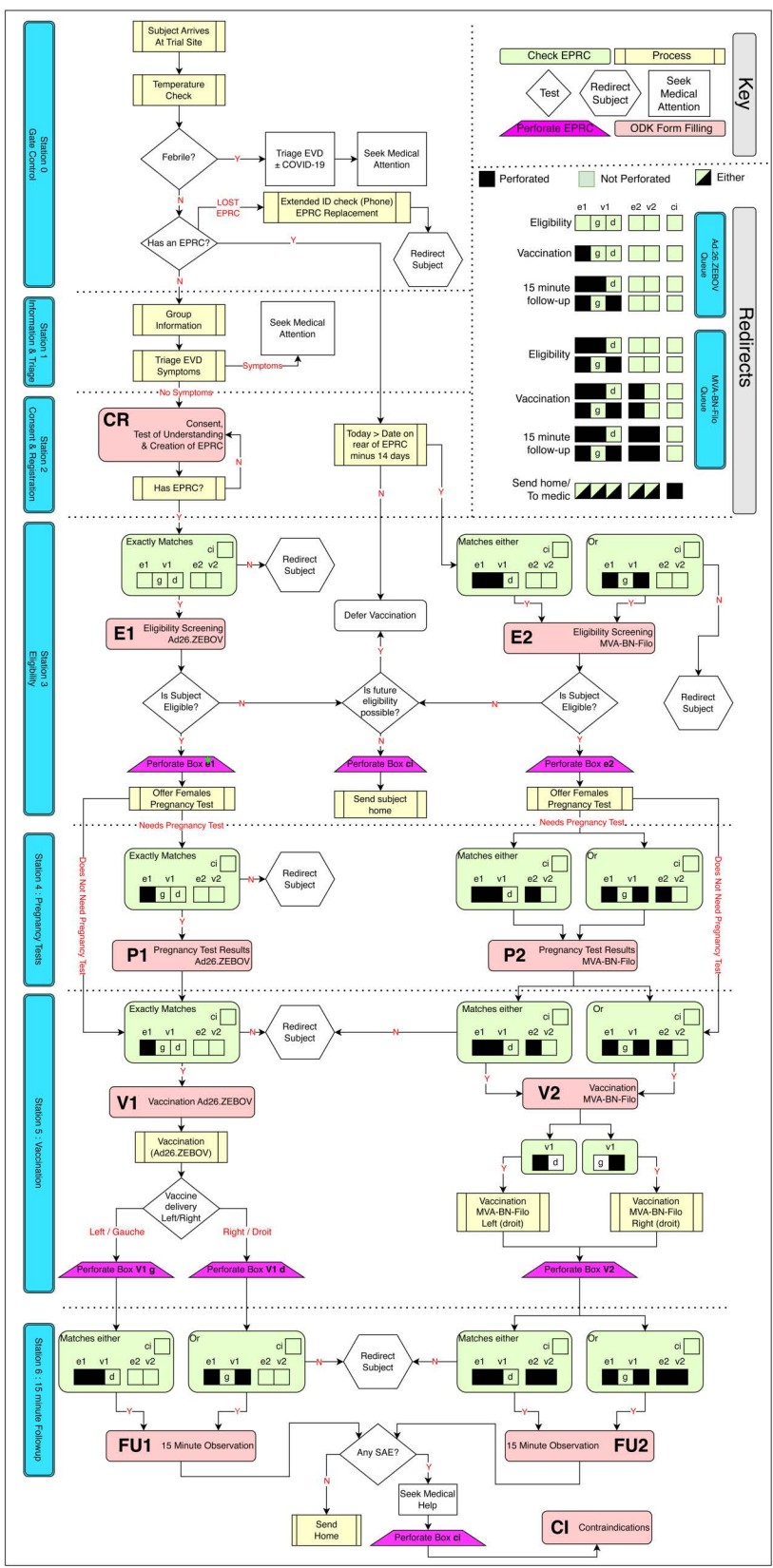

**Fig 4. Overview of the key logic in the movement of participants between stations at vaccination sites.** Electronic standard operating procedures bridge the interface between analogue and digital elements. Each eCRF contained

numerous checks and constraints, but the complex logic of individual forms is not shown here. The box labelled "Redirects" should be referred to whenever the flow-chart instructs the user to "Redirect participant". Red shading shows stages at which specific ODK forms (CR, E1, E2, P1, P2, V1, V2, FU1, FU2, CI) referred to in Figs 1 and 3 were completed.

(normally at the end of each day). To comply with data security, all raw data from ODK Aggregate were pulled and decrypted daily using ODK Briefcase (a desktop Java application) to secure file transfer protocol (sFTP) servers held by the data management team in France and the DRC. ODK Aggregate and ODK Briefcase have since been deprecated and replaced by ODK Collect.

## Study participant management system

By combining the use of the analogue EPRC and the digital ODK eCRFs/eSOPs, we created an SPMS that could manage participants and their data according to the specifications above. Every ODK form used in this system requested that the study team scan the QR/Aztec code from the sticker on the EPRC. This simple step ensured that all data related to a specific participant could be linked together through the *UniqueID* field, thereby harmonising these across time and between study sites. This instrument eliminated the need for a web-based relational database system and any need to synchronise data across either the internet or a local network (Fig 3). Where the participant was not seen in-person at the vaccination site (for example, if they contacted the study team in Goma by telephone) they were asked to read the ID number recorded on their card.

From September 2020 onwards, following the restart of the study after suspension due to the COVID-19 pandemic, participants who had lost their EPRC were able to obtain a replacement on presentation of a legal identity document and either their copy of the signed consent form or provision of correct answers to security questions, such as their date of birth and telephone number. In these cases, a telephone call to a central office allowed the field team to identify a participant's study ID from the archived data.

## Methods to reduce loss to follow-up

Several different strategies were employed to encourage participants to return for dose 2 and to reduce loss to follow-up (LTFU). All EPRCs included a 'reminder date' box which showed the date on which the second visit should take place (Fig 2). Close to the date of the second visit, attempts were made to telephone all participants who had provided a contact number during their first visit. Short messaging service (SMS) text messages were similarly sent to all participants who had provided a phone number. Those participants who were unable to provide a phone number were visited at home. All participants who reported pregnancies between the day of enrolment for dose 1 and up to 30 days after dose 2 received active follow-up by telephone call or home visit. Radio broadcasts, loudspeaker/public address announcements, posters and leaflets were also used as part of the campaign to maximise the dose 2 adherence rate.

During the eligibility screening for vaccination with dose 2, participants were asked to describe which of the various reminder approaches had prompted them to return to the study site. Study staff were asked to record 'How did the participant know when to return for the second dose of vaccine?'. The available options, from which participants could select multiple responses, were 'The date on their vaccination card', 'Radio broadcasts', 'Posters','Someone knocked on the door', 'Loudspeaker or public address announcement', 'Leaflet', 'Telephone call', and 'SMS'.

### Exit interviews

Between September and December 2020, trial participants were approached at the clinic exits and were offered the opportunity to participate in an ethnographic research project led by the trial's social sciences team. Sampling was based on convenience, with participants selected based on their availability and willingness to take part. Fourteen exit interviews were conducted. All participants provided written informed consent. The social sciences team asked trial participants 'What did you think of the photocard you were given? Was it useful and if so, which parts were useful; and if not, why not?' The conversations were carried out in Swahili and French, and were recorded, transcribed, and translated. All participants remain anonymous. The results of the exit interviews have been previously published [14].

### Analysis

Statistical and descriptive analytics to determine the utility of the date on the EPRC and other prompts to return for dose 2 among those who received the second dose were performed using R v4.1. UpSet visualisations [15] were used to describe the complex intersections and relative prominence of both individual and combinations of prompts to return for dose 2. A multivariable logistic regression model was used to investigate factors associated with participants relying on the date on the EPRC as a prompt to return for the second vaccine dose. This approach was chosen because it could determine the relative odds of participants using the EPRC as a prompt (yes/no) whilst adjusting for potential confounding factors. Covariates included participant sociodemographic, geographic proximity to vaccination sites, and health-related variables such as pre-existing chronic conditions and pregnancy status. A random intercept for study site was included to account for clustering effects due to the multi-site design. Reference groups were chosen for each categorical variable based on their relevance to the study context, typically reflecting the most common or neutral category (e.g., employed participants, adults aged 25–44 years, and females who were neither pregnant nor breastfeeding). These reference groups provided a baseline against which the effects of other categories could be compared.

## Results

Vaccination activities commenced in November 2019 and ultimately 20,723 people were screened, of whom 20,427 were eligible to take part in the study. An additional 19 participants were subsequently excluded because original copies of their consent forms could not be located, leaving 20,408 consenting participants in the final analysis, of whom all (100%) received the first vaccine dose. By the end of the vaccination period in February 2022, 15,328 (75%) participants had received both doses, although dose 2 delivery was delayed in many participants because of COVID-19 restrictions. A total of 9,280 (60.5%) participants who received dose 2 did so within the target window, whilst a further 6,043 (39.5%) received it between four and fifteen months after dose 1. Five participants (0.02%) received dose 2 less than 42 days after dose 1.

At the height of activities, the six vaccination sites were conducting almost a thousand participant visits each day using the analogue EPRC and the offline ODK data collection tool. Staff relied on the perforation system on the EPRC to move participants along the two-queue set up at the sites, with each queue being allocated to deliver one of the two different doses (either Ad26.ZEBOV or MVA-BN-Filo). The system proved to be highly efficient; out of 20,408 participants who received 35,736 doses of vaccines, none received two doses of Ad26.ZEBOV (dose 1) or the vaccine components in the wrong order. Nevertheless, the system was not perfect; one participant received dose 2 before 42 days after dose 1 and two other

participants received dose 2 twice. The two participants who incorrectly received dose 2 twice, returned to the vaccination site between 2 and 7 months following dose 2, advising that they had lost their cards. Although the data management system correctly re-identified the participants, without the original card with the perforations indicating that he had already been fully vaccinated, unfortunately, study staff did not verify their vaccination status, and they mistakenly received dose 2 for a second time. Other protocol deviations and violations included participants being vaccinated with dose 2 outside of the protocol-specified window, missing eCRFs, duplicate participant identifiers were created, discrepancies in participants' ages between dose 1 and 2, pregnancy tests requested but not performed, an erroneous pregnancy record, and participants leaving prior to the end of the 15-minute observation period following vaccination. Therefore, it should be noted that the system also required a high level of staff training and adherence to the eSOP.

Only 26 (0.13%) participants had a completed eCRF form that deviated from the study protocol (i.e., a specific eCRF was completed when it was not required) and 23 (0.11%) who received both doses were found to have one or more missing eCRFs. The missing or incomplete data included follow-up after dose 2 (n = 8), 15-minute follow-up after dose 1 (n = 4), eligibility for doses 1 (n = 3) or 2 (n = 6), consent and registration forms (n = 1), and receipt of dose 2 (n = 1).

While all participants had a vaccination card with a date reminder 37,250 of 48,432 (76.9%) SMS were successfully delivered to 16,469 participants, while only 15,153 of 53,230 (28.5%) phone calls were successfully delivered to 17,284 participants. Among the 15,327 participants who provided information about the type of prompt or reminder to return for dose 2, 8,122 (53.0%) mentioned the date reminder section of the EPRC labelled 'Please come back to the site from dd/mm/yyyy", with 5,529 (36.1%) mentioning only the EPRC, whilst the remaining 2,593 (63.9%) mentioned the EPRC in addition to at least one of the other various prompts. A total of 6,673 (43.5%) of all respondents mentioned at least one of the other reminders used in the study, but did not specifically refer to the EPRC's date reminder, while 532 (3.48%) did not mention any type of prompt (Fig 5). The most frequently cited factors that prompted a return for dose 2 were thus the EPRC, a home visit (n=5,321, 34.7%) and phone calls (n=4,698, 30.7%). SMS (n=1,289, 8.4%) and other prompts (n=2,018, 13.2%) appeared to have had comparatively lower impacts.

Compared to the reference age group of 25–44 years, those aged 44–64 years were more likely to use the date on the card (OR=1.19, 95%CI 1.00-1.41, p=0.047) while those in younger ages groups were less likely (0–11 years: OR=0.47 (95%CI 0.38-0.59, p<0.001); 12–17 years: OR=0.61 (95%CI 0.46-1.10, p<0.001); and 18–24 years: OR=0.78 (0.68-0.90, p=0.001).) Those who were aged under 18 years or students aged 18 years and over were more likely to have used the date on the EPRC compared to those who were employed (OR=1.61, 95%CI 1.27-2.04, p<0.001 and OR=1.30, 95%CI 1.01-1.68, p=0.042, respectively) and those who lived outside the health area of the vaccination site were less likely (OR=0.62, 95%CI 0.55-0.69, p<0.001). Compared to females who were neither pregnant nor breastfeeding, those who were breastfeeding were less likely to have used the date (OR=0.78, 95%CI 0.61-0.98, p=0.033). However, those who were pregnant were neither more nor less likely to have used it (OR=1.06, 95%CI 0.82-1.36, p=0.665). Those for whom the COVID-19 lockdown coincided with the scheduled date of dose 2 were less likely to have used the date (OR=0.83, 95%CI 0.72-0.96, p=0.011) and those who mentioned a home visit, telephone call, SMS or other prompt as a reminder to return for dose 2, were less likely to have relied on it (home visit: OR=$7\times10^{-3}$ (95%CI $6\times10^{-3}$ - $8\times10^{-3}$, p<0.001); telephone call: OR=0.17 (95%CI 0.15-0.19), p<0.001); SMS (OR=0.17 (95%CI 0.14-0.21, p<0.001); and other (OR=0.54 (95%CI 0.46-0.65), p<0.001). Those for whom a pre-existing and chronic health condition was identified at the second visit were more likely to have used the date on the ERPC (OR=1.13, 95% CI 1.10-1.56, p=0.002). However, there was no association with those who had a condition identified at the first

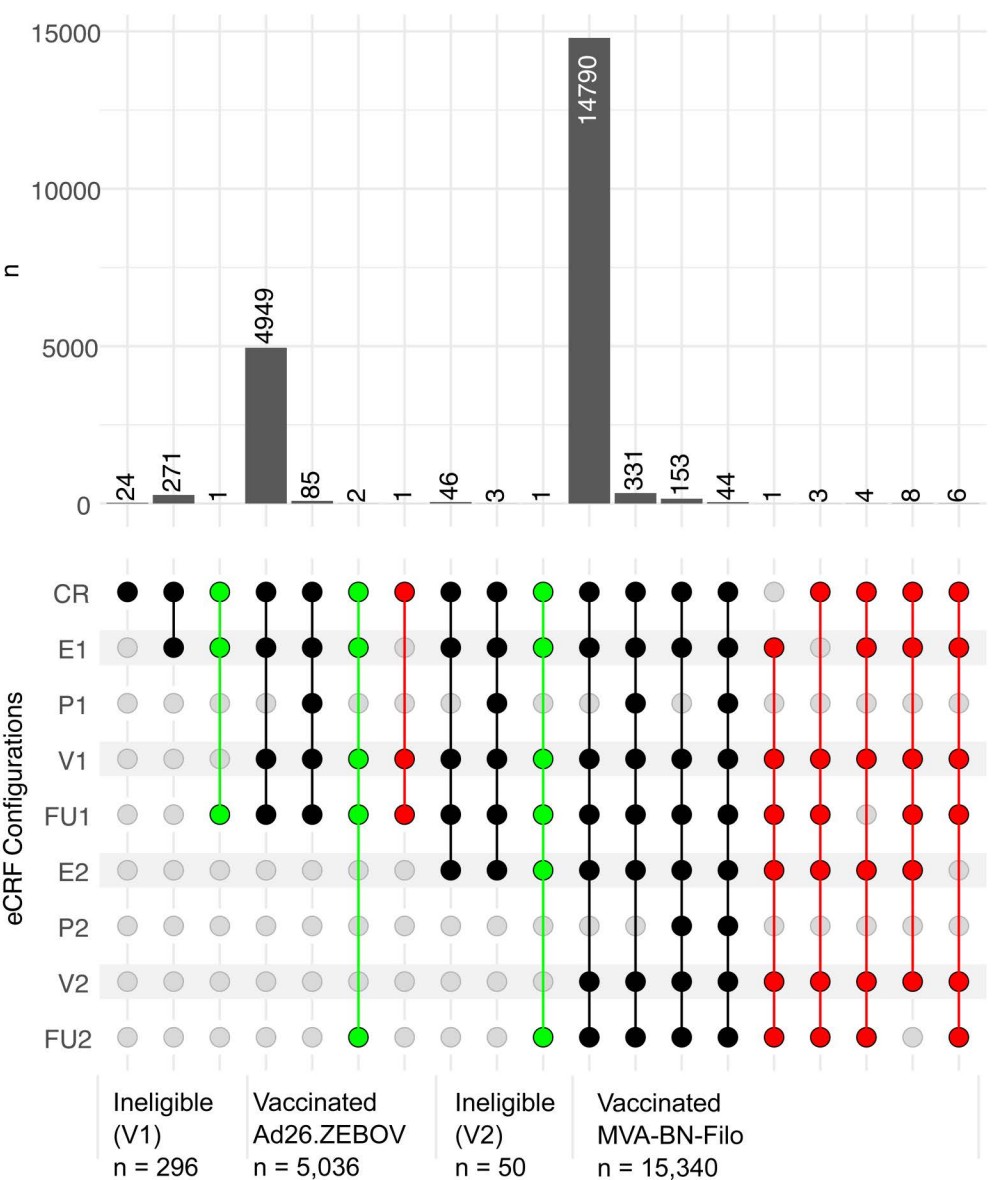

**Fig 5. Factors identified as having prompted a return for dose 2.** The vertical bars represent the number of participants for whom each combination of reminders prompted them to return for dose 2 and the dots represent the combination. The horizontal bars show the total number of participants who mentioned the prompt.

visit (OR=0.94, 95% CI 0.81-1.10, p=0.452) (Table 1). The participants of the trial reported a broad range of conditions, which can be summarised as: 1) **Infections**, including sexually transmitted infections, tuberculosis, herpes and others; 2) **Musculoskeletal and joint issues**, especially arthropathies, rheumatoid arthritis, fractures and soft tissue injuries; 3) **Gastrointestinal issues**, including inflammatory bowel disease, irritable bowel syndrome, gastric issues and ulcers; 4) **Neurological and mental health conditions** including schizophrenia, bipolar disorder, epilepsy, migraine and seizures; 5) **Cardiovascular and respiratory issues**, including hypertension, diabetes, heart disease, asthma and pneumonia-like symptoms; 6) **Skin and allergy conditions** such as psoriasis, eczema and allergies; 7) **Trauma and substance-related issues** including burn injuries, substance abuse and narcotic addiction.

**Table 1. Demographic features of the trial and logistic regression model of factors associated with reporting the date written on the Enhanced Participant Record Card (EPRC) as a prompt for returning for dose 2, with study site as a random effect.**

| Variable | Participants | Cited EPRC date as a prompt to return for dose 2 | Odds ratio (95% CI) | p value |
|---|---|---|---|---|
| | n (% of 15,327) | n (% of group total) | | |
| **Study Site** | | | | |
| 101 | 2,882 (18.8%) | 1,011 (35.1%) | – | – |
| 102 | 2,227 (14.5%) | 1,198 (53.8%) | – | – |
| 103 | 2,638 (17.2%) | 1,364 (51.7%) | – | – |
| 104 | 2,546 (16.6%) | 1,445 (56.8%) | – | – |
| 105 | 2,645 (17.3%) | 1,756 (66.4%) | – | – |
| 106 | 2,389 (15.6%) | 1,348 (56.4%) | – | – |
| **Employment status** | | | | |
| Employed worker (reference) | 6,182 (40.3%) | 3,609 (58.4%) | – | – |
| Unemployed | 4,663 (30.4%) | 2,276 (48.8%) | 1.10 (0.96-1.26) | 0.181 |
| Student (aged over 18 years) | 625 (4.1%) | 385 (61.6%) | 1.30 (1.01-1.68) | 0.042 |
| Under 18 years | 3,857 (25.2%) | 1,852 (48.0%) | 1.61 (1.27-2.04) | <0.001 |
| **Required more than one attempt at test of understanding** | 733 (4.8%) | 414 (56.5%) | 0.98 (0.78-1.22) | 0.825 |
| **Lives outside the health area of the vaccination site** | 4,218 (27.5%) | 1,823 (43.2%) | 0.62 (0.55-0.69) | <0.001 |
| **Gender, pregnancy, and breastfeeding** | | | | |
| Female (reference) | 6,292 (41.1%) | 3,214 (51.1%) | – | – |
| Female and pregnant | 644 (4.2%) | 325 (50.5%) | 1.06 (0.82-1.36) | 0.665 |
| Female and breastfeeding | 764 (5.0%) | 354 (46.3%) | 0.78 (0.61-0.98) | 0.033 |
| Male | 7,627 (49.7%) | 4,229 (55.4%) | 1.03 (0.93-1.15) | 0.538 |
| **Age (years)** | | | | |
| 0 – 11 | 3,564 (23.3%) | 1,461 (41.0%) | 0.47 (0.38-0.59) | <0.001 |
| 12 - 17 | 1,704 (11.1%) | 885 (51.9%) | 0.61 (0.46-1.10) | <0.001 |
| 18 - 24 | 3,361 (21.9%) | 1,809 (53.8%) | 0.78 (0.68-0.90) | 0.001 |
| 25 – 44 (reference) | 4,532 (29.6%) | 2,595 (57.3%) | – | – |
| 45 - 64 | 1,821 (11.9%) | 1,152 (63.3%) | 1.19 (1.00-1.41) | 0.047 |
| 65 and over | 345 (2.3%) | 220 (63.8%) | 1.09 (0.76-1.57) | 0.642 |
| **COVID lockdown coincided with scheduled date of dose 2** | 12,419 (81.0%) | 5,972 (48.1%) | 0.83 (0.72-0.96) | 0.011 |
| **Pre-existing and chronic health conditions (identified at the first visit)** | 2,240 (14.6%) | 1,315 (58.7%) | 0.94 (0.81-1.10) | 0.452 |
| **Pre-existing and chronic health conditions (identified at the second visit)** | 1,663 (10.9%) | 1,013 (60.9%) | 1.31 (1.10-1.56) | 0.002 |
| **Reminder which prompted to return for dose 2** | | | | |
| Home visit | 5,321 (34.7%) | 563 (10.6%) | $7 \times 10^{-3}$ ($6 \times 10^{-3}$–$8 \times 10^{-3}$) | <0.001 |
| Telephone call | 4,704 (30.7%) | 2,013 (42.8%) | 0.17 (0.15-0.19) | <0.001 |
| SMS | 1,289 (8.4%) | 205 (15.9%) | 0.17 (0.14-0.21) | <0.001 |
| Other* | 2,018 (13.2%) | 462 (22.9%) | 0.54 (0.46-0.65) | <0.001 |

*Other prompts included any combination of radio or loudspeaker announcements, posters or leaflets.

## Discussion

Running GCP-compliant clinical trials in areas with unreliable or limited infrastructure is challenging. In a previous study in Sierra Leone, Carter and colleagues invested in building a reliable internet connection and web-based electronic CDMS, but had to return to paper-based data entry forms when the connectivity remained unreliable [4]. Although other novel

CDMS approaches were deployed during both the West Africa Ebola epidemic (2014–2016) and as part of a vaccine trial in western DRC in 2019–2022, our system allowed data to be synchronised in almost real-time to a central server, thus avoiding the need for a local server, a key feature of the previously described systems [5–7]. Furthermore, by combining the use of EPRCs with the off-grid capable electronic data collection system ODK [12], we were able to facilitate the recruitment, management and monitoring of 20,408 participants as they progressed through the DRC-EB-001 trial, even during periods when there was no internet connection. Whilst study participant cards are a common feature of trials, typically acting as both a form of identification and a prompt to return for future appointments [4], our system used the card as a fully integrated and critical component of the data platform in a large-scale longitudinal study.

It was encouraging that the vast majority (14,665, 95.7%) of participants who returned for dose 2 had kept their EPRC, while an additional 663 (4.3%) were successfully re-identified and vaccinated after declaring their EPRC lost. The EPRC appeared to serve as an effective prompt for participants to return for dose 2, having been mentioned by more than half, and the only prompt cited by over one-third. Of course, we used a range of reminders in the study, for which the specific impact of each cannot be ascertained; in addition to the EPRC, home visits and phone calls may have made significant contributions. A randomized controlled trial (RCT) conducted in Kenya reported that individuals who received an SMS and phone-call or in-person reminder were 1.4 times as likely to return for repeat HIV testing than individuals who received only an appointment card [16]. During a separate RCT conducted in Tanzania, no difference in attendance at 14-month follow-up for cervical cancer screening was found among those who received a text message reminder compared to an appointment card which was the standard of care [17]. Although we did not perform similar analyses, those who used a prompt other than the date on the EPRC as a reminder to return for dose 2 were less likely to use the EPRC as a reminder. Nevertheless, whilst more complex communication methods may be effective (and cost-effective) in some settings, simple methods such as the EPRC with personalised communication may be more effective, as indicated by the finding that considerably fewer of our study participants appeared to respond to SMS message prompts that were sent to all participants who provided a phone number.

We were unsurprised that the EPRC was less likely to act as reminder of the date to return in the younger age groups, potentially because some of the children were included in a safety subset who received active follow-up via telephone calls from the study team. It is also possible that if children were vaccinated at the same time as their parents, the parents used the date on their own EPRC as a prompt for the family to return for dose 2. Similarly, the same reasoning might be applied to those who were breastfeeding and actively followed up, despite the effect being smaller. Interestingly, we did not see the same effect among pregnant women, even though some of them were also being actively followed up. The reason for this needs to be further explored. It is possible that those aged 18–24 years who are more familiar with technology were reliant on other methods such as SMS reminders or stored the date in their phones. However, if this was the case we might have expected students to also be less reliant on the EPRC. For reasons which remain unclear, those who lived outside the health area of the vaccination site were less likely to use the data on the EPRC. It is possible that some of these participants were traders and more likely to be reminded about dose 2 through community engagement activities. Also of interest was the positive association between the reporting of pre-existing and chronic health conditions during the return for dose 2 and the use of the EPRC but the absence of this association during the initial visit for dose 1. This may suggest that those with a newly diagnosed chronic condition may be more attentive to appointment

reminders. Although it may not be seen as surprising that those for whom the COVID-19 lockdown coincided with the scheduled date of dose 2 were less likely to refer to the date on the EPRC, given that the date became irrelevant, it should be noted that the odds ratio was only 0.83, and hence several participants were still relying on this. This may suggest that these participants were conscious that being 'late' for dose 2 acted as a prompt to return once the study resumed.

Other unintended and unanticipated factors may have influenced participants' willingness to be vaccinated; previously published ethnographic research in Goma revealed how the EPRCs used in this study were well received by the participants because their possession gave the holder a sense of legitimacy and franchise, proving their identity and ensuring access to both the second vaccine and medical care (although, in fact, the card was not required to access care) [14]. Over a third of participants expressed value of the study phone number on the card (which was used more than 15,000 times during the study) as a personal and direct line of contact with the study team. In addition, some participants considered the EPRCs to hold socio-political value beyond the context of the vaccine study, affording advantages for travel both within the DRC and to cross borders, effectively constituting an "Ebola vaccine passport" [14], although there was, in fact, never any requirement for EVD vaccination related to mobility during the EVD outbreak. Nevertheless, considering that the study took place in part during the COVID-19 pandemic, when travel restrictions and requirements for COVID-19 vaccine certificates were being introduced (in addition to pre-existing requirements in some circumstances to show proof of yellow fever vaccination), it is understandable that participants may have perceived this advantage of the EPRC. Future studies should consider how identification systems for clinical trials might be understood locally in relation to nationally recognised or regionally implemented health passports, routine immunisation records, and regarding socio-political dynamics.

In addition to the utility of the EPRC as a prompt to return for dose 2, it had benefit for quality control of the SPMS during the study; missing eCRFs or those for whom completion did not comply with the study protocol were very few (just over 0.1%). While we suspect that the prompts built into the eCRF to check the EPRC contributed to this, we also acknowledge the importance and potential influence of the training and monitoring of study staff. If a similar SPMS is used for a study of this nature, we recommend obtaining feedback from staff to understand the benefits of these prompts. The EPRCs also provided a mechanism by which participants could be passively followed up for SAEs and notification of pregnancies. This was also helpful when there was a significant migration of participants out of Goma during the volcanic eruption in May 2021.

The accurate identification of participants during follow-up visits in longitudinal clinical trials is an important aspect of participant safety [18]. Due to a history of insecurity and mistrust in North Kivu, as well as the lack of internet connectivity, we decided to identify participants using analogue photo EPRC instead of digital biometric solutions such as iris and fingerprint scanning. Whilst the use of photographs on the EPRC provided a practical and effective basis on which the study team could confirm an individual's identity, they did not provide verifiable proof that the individuals who were vaccinated at visit one were the same individuals who were vaccinated at dose 2 under the same *unique ID*. Biometrics such as fingerprint or iris scanning are increasingly used as an alternative method for identification of study participants and for routine medical care [19,20]. In a similar context, iris scanning was used during the EBL2007 trial in Boende to determine participants' identity [7]. If acceptable to the target community, future studies may consider implementing a quantitative biometric tool such as the novel ODK fingerprint capture Application Program Interface that we recently described [21] or digital fingerprints as used in a trial of human papilloma virus

vaccines in Tanzania [22]. This tool retains the off-grid functionality of ODK, allowing forms to capture ISO/IEC 19794–2 fingerprint templates within each eCRF. Software checks could then be used to compare fingerprints and provide biometric confirmation of identities and biometric linkage between eCRF data collected at different timepoints.

Despite the apparent success of the hybrid analogue-digital database in this study, we acknowledge some limitations to the work. Given the operational context of the DRC-EB-001 trial, which was rapidly implemented due to the nature of EVD epidemic and the need to trial vaccines, and took place during two separate World Health Organisation-declared Public Health Emergencies of International Concern (the EVD epidemic and the COVID-19 pandemic), we were unable to perform a robust side-by-side performance evaluation of the system against any other type of SPMS, which would have provided an opportunistic analysis. As such, our evaluations are limited to mostly observational and qualitative findings. However, this is an advancement of the implementation of the electronic tool described by Jobanputra et al., 2017 who noted that it was delivered too late to determine any improvement in the efficiency of the EMCs [5]. To enhance the utility of the EPRC in regard to the safety of the participants in future trials conducted in sites without access to a digital central database, we plan to more fully develop methods to record contraindications which may occur away from the study site where the perforator tools are available, and a system by which (possibly coded) information could be added to an EPRC by the participant, acting under instructions given over the telephone. We also plan to include more information on the EPRC itself to increase the granularity of data in the analogue database. For instance, additional perforations could help to record evidence from *ad hoc* site visits or to add more complex or text-based data entities. We may also expand the function of the EPRC to include numeric, categorical, time-series and date-time entities and to involve the use of low-cost/low-power digital devices. Although the proportion of EPRCs lost was low, we acknowledge that allowing participants to keep their vaccination cards may be a limitation. While storing the EPRCs with the trial team may have been safer, given that participants could present to any vaccination site or healthcare centre, this may have presented logistical challenges. Lastly, although the numbers of protocol deviations and violations were low, to avoid participants being vaccinated too early, we could consider implementing cross-checks of eligibility using registration books at clinics, whilst bearing in mind the limitations of these when the number of trial participants is large and there are multiple sites.

## Conclusion

We demonstrate how the combination of an off-grid capable electronic data collection software and decentralised analogue data stored on enhanced participant study cards was able to facilitate effective and robust participant and electronic data management in a large-scale vaccine study conducted in the context of a complex disease outbreak considered a public health emergency. Less than 1% of the eCRFs deviated from the study protocol and, furthermore, over 95% of participants who returned for dose 2 kept their vaccination card, with more than half using the date on the card as a prompt to return. By remaining fully functional across time and geographical space without reliance on consistent access to the internet and/or reliable supply of electricity, the system fills a methodological gap for complex trials taking place in areas with limited infrastructure. However, we acknowledge challenges with the system, including the inability of the EPRC to record events and contraindications that occurred in locations other than the vaccination site and, therefore, we would recommend developing a system to incorporate these in the future. Additionally, if this mechanism is to be used in the future alongside other prompts to return for vaccination doses, such as SMS, we would recommend designing a study to robustly compare their performance.

## Acknowledgements

We thank all members of the study team and the study participants. We also thank World Vision and MSF for their support with community engagement activities and Babajide Keshinro from Johnson & Johnson Innovative Medicine for his support with data management.

## Author contributions

**Conceptualization:** Hannah E. Brindle, Chrissy h. Roberts.

**Data curation:** Darius Tetsa-Tata, Chrissy h. Roberts.

**Formal analysis:** Hannah E. Brindle, Tansy Edwards, Chrissy h. Roberts.

**Funding acquisition:** Daniel G. Bausch, Jean-Jacques Muyembe-Tamfum, Deborah Watson-Jones.

**Methodology:** Hannah E. Brindle, Chrissy h. Roberts.

**Software:** Darius Tetsa-Tata, Ibrahim Seyni Ama, Anton Camacho.

**Visualization:** Chrissy h. Roberts.

**Writing – original draft:** Hannah E. Brindle, Chrissy h. Roberts.

**Writing – review & editing:** Hannah E. Brindle, Darius Tetsa-Tata, Tansy Edwards, Edward Man-Lik Choi, Kambale Kasonia, Soumah Aboubacar, Grace Mambula, Hugo Kavunga-Membo, Rebecca Grais, John Johnson, Daniel G. Bausch, Jean-Jacques Muyembe-Tamfum, Ibrahim Seyni Ama, Shelley Lees, Anton Camacho, Deborah Watson-Jones, Chrissy h. Roberts.

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
