## [Decision Letter · Decision Letter 0]

31 Oct 2024

PGPH-D-24-01648

An analogue-digital hybrid clinical data management platform for use in resource limited settings: Evidence from a two-dose preventive Ebola virus vaccine trial in the Democratic Republic of the Congo

Dear Dr. Brindle,

Thank you for submitting your manuscript to PLOS Global Public Health. After careful consideration, we feel that it has merit but does not fully meet PLOS Global Public Health’s publication criteria as it currently stands. Therefore, we invite you to submit a revised version of the manuscript that addresses the points raised during the review process.

The manuscript has been assessed by two reviewers and their comments are available below. They have requested some revisions including improved reporting of statistical methods among other points. Please review their comments and make the appropriate revisions.

We look forward to receiving your revised manuscript.

Kind regards,

Emma Campbell, Ph.D

Staff Editor

Journal Requirements:

Reviewers' comments:

Reviewer's Responses to Questions

**Comments to the Author**

1. Does this manuscript meet PLOS Global Public Health’s publication criteria ? Is the manuscript technically sound, and do the data support the conclusions? The manuscript must describe methodologically and ethically rigorous research with conclusions that are appropriately drawn based on the data presented.

Reviewer #1: Yes

Reviewer #2: Yes

2. Has the statistical analysis been performed appropriately and rigorously?

Reviewer #1: Yes

Reviewer #2: Yes

3. Have the authors made all data underlying the findings in their manuscript fully available (please refer to the Data Availability Statement at the start of the manuscript PDF file)?

Reviewer #1: No

Reviewer #2: Yes

4. Is the manuscript presented in an intelligible fashion and written in standard English?

Reviewer #1: Yes

Reviewer #2: No

5. Review Comments to the Author

Reviewer #1: Congratulations on tackling this very important topic on data collection in LMICs. This is indeed a challenging topic and I read this paper with great interest on a possible solution you developed.

I have some minimal question for clarification:

Line 61: Please go into a little more detail on the types of chronic conditions that were collected. How could these conditions influence the attendance to the clinics for vaccination?

Line 178-179: Could you elaborate briefly why the pregnancy test was optional or why it is relevant information for this paper?

Line 358-361: Can you clarify which data was collected electronically and at what time point? Did the participant who received the second dose twice come to the vaccination site twice on the same day, and was the electronic data collection tool therefore not updated yet?

Line 365: can you elaborate on what you mean by "eCRF deviated from the study protcol"?

General: Is it possible to explain what data was collected in the eCRF? It is unclear whether more information than what was available on the perforated cards was collected, and if so, how and when this data became available on the central server.

Reviewer #2: I have thoroughly read through the paper, and I have the following comments for the authors.

1. The title of the paper is too long and contain an excessive explanation of the study. Research article titles should be very concise and communicate the theme of the study. Kindly reconstruct the title.

2. The novelty of the study is not clearly captured in the introduction. I expect the authors to carry out an extensive literature study on the existing clinical data management system (CDMS). This will enable the reader to identify the gaps in the existing system and appreciate the contribution to literature for this study. Please carry out a good literature review and be clear with the novelty of the study.

3. The authors should clearly state with reasons the statistical designs applied in this research. For instance, be clear with the research design, sampling technique and other salient techniques.

4. The arrangement format of the Figures in the study is inappropriate. You may create an appendix and send both the figures and their titles there or let them appear in the main work together. Titles for figures cannot appear in the work while the figures appear later.

5. The study results are quite questionable since I see no proper comparison with existing results. Authors are encouraged to compare their results to that of the existing approaches. This will help verify the validity of their results.

6. The paper contains enormous grammatical errors. Please fix them.

7. I have challenge with the contribution of this research work, since the figures are poorly presented. I am unable to read the patterns in there to assess its feasibility.

8. Authors may employ more statistical tools like graphs to present the results in the table. That will make reading quite interesting.

9. The limitation of this current study is enormous and questions what problem exactly has been solved by this study.

10. Authors should work on the in-text citations. Check the journal referencing formats and follow appropriately.

6. PLOS authors have the option to publish the peer review history of their article (what does this mean? ). If published, this will include your full peer review and any attached files.

**Do you want your identity to be public for this peer review?** For information about this choice, including consent withdrawal, please see our Privacy Policy .

Reviewer #1: No

Reviewer #2: No

---

## [Decision Letter · Decision Letter 1]

5 Mar 2025

PGPH-D-24-01648R1

Using an analogue-digital hybrid clinical data management platform during a two-dose preventive Ebola virus vaccine trial in Goma, the Democratic Republic of the Congo

Dear Dr. Brindle,

Thank you for submitting your manuscript to PLOS Global Public Health. After careful consideration, we feel that it has merit but does not fully meet PLOS Global Public Health’s publication criteria as it currently stands. Therefore, we invite you to submit a revised version of the manuscript that addresses the points raised during the review process.

We look forward to receiving your revised manuscript.

Kind regards,

Julio Croda, Ph.D, M.D.

Academic Editor

Journal Requirements:

Additional Editor Comments (if provided):

Reviewers' comments:

Reviewer's Responses to Questions

**Comments to the Author**

1. If the authors have adequately addressed your comments raised in a previous round of review and you feel that this manuscript is now acceptable for publication, you may indicate that here to bypass the “Comments to the Author” section, enter your conflict of interest statement in the “Confidential to Editor” section, and submit your "Accept" recommendation.

Reviewer #1: All comments have been addressed

Reviewer #2: All comments have been addressed

2. Does this manuscript meet PLOS Global Public Health’s publication criteria ? Is the manuscript technically sound, and do the data support the conclusions? The manuscript must describe methodologically and ethically rigorous research with conclusions that are appropriately drawn based on the data presented.

Reviewer #1: Yes

Reviewer #2: Partly

3. Has the statistical analysis been performed appropriately and rigorously?

Reviewer #1: Yes

Reviewer #2: Yes

4. Have the authors made all data underlying the findings in their manuscript fully available (please refer to the Data Availability Statement at the start of the manuscript PDF file)?

Reviewer #1: Yes

Reviewer #2: Yes

5. Is the manuscript presented in an intelligible fashion and written in standard English?

Reviewer #1: Yes

Reviewer #2: Yes

6. Review Comments to the Author

Reviewer #1: 1. Thank you for adding the pregnancy clarifcation, the reasoning is now clear. Do I understand correctly that pregnancy was thus not an exclusion criteria for vaccination? I believe some punctuation is missing in the added text. If this is corrected, I have no further comments on this point. Please also be consistent in refering to the doses: Dose one versus Dose 2 in one sentence. Propose to write it out in text or in letters but not the combination.

Reviewer #2: I have gone through the revised manuscript and find he results to be interesting. I have some few comments which will improve the paper.

1. The abstract should be a paragraph and should be concise.

2. There are still some errors in the manuscript, for instance, check line 67 on page 2.

3. The conclusion is not accurately written. key observations from the research were not reported. Also, it fails to state the weakness of the research and also recommendations for future.

4. The figure titles are too broad. I suggest that concise titles should be given, and the details should be given under the discussion of results.

7. PLOS authors have the option to publish the peer review history of their article (what does this mean? ). If published, this will include your full peer review and any attached files.

**Do you want your identity to be public for this peer review?** For information about this choice, including consent withdrawal, please see our Privacy Policy .

Reviewer #1: No

Reviewer #2: No

---

## [Editor Report · Decision Letter 2]

18 Mar 2025

Using an analogue-digital hybrid clinical data management platform during a two-dose preventive Ebola virus vaccine trial in Goma, the Democratic Republic of the Congo

PGPH-D-24-01648R2

Dear Dr Brindle,

We are pleased to inform you that your manuscript 'Using an analogue-digital hybrid clinical data management platform during a two-dose preventive Ebola virus vaccine trial in Goma, the Democratic Republic of the Congo' has been provisionally accepted for publication in PLOS Global Public Health.

Best regards,

Julio Croda, Ph.D, M.D.

Academic Editor